# Impact of fatty acid composition on markers of exocrine pancreatic stimulation in dogs

**Yunyi Zhang**[1], **Claudia A. Kirk**[1], **M. Katherine Tolbert**[2], **Jörg M. Steiner**[2], **Dallas Donohoe**[3], **Maryanne Murphy**[1], **Cary Springer**[4], **Angela Witzel-Rollins**[1]*

1 Department of Small Animal Clinical Sciences, The University of Tennessee College of Veterinary Medicine, The University of Tennessee, Knoxville, Tennessee, United States of America, 2 Gastrointestinal Laboratory, Department of Small Animal Clinical Sciences, School of Veterinary Medicine and Biomedical Sciences, Texas A&M University, College Station, Texas, United States of America, 3 Department of Nutrition, The University of Tennessee, Knoxville, Tennessee, United States of America, 4 Research Computing Support, Office of Information Technology, The University of Tennessee, Knoxville, Tennessee, United States of America

* arollins@utk.edu

**Data Availability Statement:** All relevant data are within the paper and its Supporting Information files.

## Abstract

Chronic pancreatitis in dogs is typically managed with a low-fat diet. Human research suggests that consumption of medium-chain triglycerides (MCT) may lessen pancreatic enzyme release compared to consumption of long-chain fatty acids (LCFA). Twelve healthy adult colony dogs were fed a meal of cod and rice with either 3% metabolizable energy (ME) fat (control), high MCT (25% ME MCT oil, 25% ME butter), high saturated LCFA (50% ME butter), or high unsaturated LCFA (50% ME canola oil) in a 4-period by 4-treatment crossover design. Serum concentrations of canine pancreatic lipase immunoreactivity, gastrin, cholesterol, triglycerides, and serum activities of amylase and DGGR lipase (1,2-*o*-dilauryl-*rac*-glycero-3-glutaric acid-(69-methylresorufin) ester lipase) were measured at times 0 (fasted), 30, 120 and 180 minutes post-prandially. Following a 3-or 4-day wash-out period, each dog was assigned a new diet and the process was repeated for all treatments. Data were analyzed as a repeated-measures mixed model ANOVA. Post-hoc pairwise comparisons were run using Tukey-Kramer adjusted p-values. Shapiro-Wilk tests were used to evaluate residual normality. All statistical assumptions were sufficiently met. Statistical significance was defined as P<0.05. Of the markers tested, only serum triglyceride concentrations were affected by treatment, with consumption of high MCT resulting in lower triglycerides than both LCFA groups at times 120 and 180 minutes (P<0.0001). As expected, the high MCT group had higher triglycerides compared to the control group (P<0.0001). The type of dietary fat consumed had little acute impact on most markers of exocrine pancreatic stimulation in healthy dogs.

## Introduction

Pancreatitis is a common disease affecting dogs. This disease is characterized by the aberrant release of pancreatic lipase and protease enzymes within pancreatic structures, which leads to

**Funding:** Our study was funded through an intramural grant from the University of Tennessee. The funders had no role in study design, data collection and analysis, decision to publish, or preparation of the manuscript.

**Competing interests:** The authors have declared that no competing interests exist.

inflammation and destruction of the pancreas. Dietary intake of fat and protein stimulates the release of several gastric and intestinal hormones that trigger the release of pancreatic enzymes for nutrient digestion. To minimize pancreatic stimulation, nutrition plans for dogs with chronic pancreatitis typically involve feeding a low-fat, moderate-protein diet [1]. Long-term use of low-fat diets can present challenges for some patients. Because fat is more calorically dense than protein and carbohydrates, low-fat diets often have a low caloric density. Dietary fat and protein are also highly palatable to dogs and formulating diets that are moderate in protein and low in fat may reduce voluntary food intake [2, 3]. Consequently, some patients with chronic pancreatitis may struggle to maintain their ideal body condition. Understanding the effect of various fat sources on pancreatic stimulation could provide better dietary management of canine pancreatitis by increasing the caloric density of foods for dogs with chronic pancreatitis.

The release of digestive enzymes from the pancreas is a multifactorial process involving neural, endocrine, and paracrine stimulation [4]. There are several tests available that can provide insight into the role of diet on enzyme release from the pancreas. Canine-specific pancreatic lipase (cPLI) is a serum marker that reflects the quantity of pancreatic lipases released into circulation and is often used to diagnose active pancreatitis in dogs [5]. Lipase may originate from several tissues in the dog (e.g. gastric, hepatic, pancreatic, etc.) with a majority released from the pancreas. As such, the measurement of pancreatic specific lipase (cPLI) provides an accurate assessment of pancreatic activity. DGGR lipase [1,2-*o*-dilauryl-*rac*-glycero-3-glutaric acid-(69-methylresorufin) ester (DGGR)] is a catalytic assay for measuring lipases [6]. This method measures the hydrolysis of substrate by potential lipases and uses colorimetric reaction to detect enzyme activity [7]. Amylase is a cytoplasmic enzyme that is synthesized and secreted by the intestines and the pancreas in dogs. Amylase is primarily released from the pancreas and hydrolyses the starch in the intestine during digestion. Finally, gastrin is a hormone released by gastric and intestinal enterochromaffin cells in response to gastric distension and increased duodenal pH. During the gastric phase of exocrine pancreatic secretion, gastrin is released from the stomach in response to protein ingestion and stimulates pancreatic enzyme secretion [8]. Measuring these substances in healthy dogs following a meal may provide insight into how fatty acid structure influences the exocrine pancreas and potentially contributes to the development of pancreatitis.

Triglycerides make up the majority of dietary fat and are comprised of a glycerol backbone bound to three fatty acids. The length of fatty acids varies, but most triglycerides contain long-chain fatty acids (LCFA) with at least 14 carbons, and LCFA are potent stimulators of pancreatic enzyme release [9]. Long-chain fatty acids can be saturated (no double bonds between carbons) or unsaturated. Fatty acids with more than one double body between carbons are termed polyunsaturated fatty acids (PUFA). Medium-chain triglycerides (MCT) contain shorter chain fatty acids with six to 12 carbon atoms. Due to their small size and lipophilic properties, some MCT can be rapidly absorbed into enterocytes and distributed via the portal circulation [10].

Research in humans suggests that MCT do not stimulate the exocrine pancreas to the same degree as LCFA [11]. In a study of eight healthy adult humans, infusion of MCT into the stomach did not significantly increase amylase, lipase, or bilirubin within the duodenum. In contrast, LCFA significantly increased the release of all three substances. In the same study, plasma cholecystokinin (CCK) was significantly higher than baseline with the LCFA administration, while MCT had no effect on CCK concentrations [11]. In addition to chain length, fatty acid saturation status may also influence pancreatic enzyme release. A human study conducted by Beardshall [12] suggested that the consumption of monounsaturated fatty acids (MUFA) and polyunsaturated fatty acids (PUFA) resulted in greater secretion of CCK

compared to the consumption of saturated fatty acids. High levels of unsaturated fatty acids, but not saturated fatty acids, also induced intra-cellular trypsin activation and cell damage of pancreatic acinar cells in vitro [13]. There is little research in dogs assessing the impact of fatty acid composition on the exocrine pancreas. A study of 10 healthy research dogs found dietary fat content to have no significant impact on serum concentrations of canine trypsin-like immunoreactivity (cTLI), cPLI, or gastrin [14]. Treatment groups consisted of dogs receiving an average-fat commercial diet, a low-fat commercial diet, an average fat diet with pancreatic enzymes, and a low-fat diet with pancreatic enzymes and MCT. Unfortunately, the fatty acid composition of the commercial test diets was not reported, and the group with added MCT also had pancreatic enzyme supplementation, confounding the role of MCT alone. While the treatment group with MCT did have numerically lower serum cPLI concentrations, this finding was not different statistically (P = 0.2).

The primary goal of this study was to evaluate markers of exocrine pancreatic stimulation in dogs eating a high MCT diet compared to a typical LCFA diet. Additionally, this study assessed the role of fatty acid saturation in pancreatic stimulation by comparing diets high in unsaturated and saturated fatty acids. Post-prandial concentrations of serum cPLI in healthy dogs were also documented as a final aim of this study.

## Materials and methods

### Experimental design

Utilizing a four-period by four-treatment crossover design, this study compared the concentrations of serum CCK, gastrin, amylase, triglycerides, cholesterol, DGGR lipase and cPLI in 12 healthy, adult research dogs consuming meals with varying fatty acid content and composition. Four dietary treatment groups were studied, consisting of a group receiving a control diet with minimal fat, one receiving a high long chain saturated fatty acid (LCSFA) diet with approximately 50% of metabolizable energy (ME) from butter, a high long chain unsaturated fatty acid (LCUFA) diet with approximately 50% ME from canola oil, and a high MCT diet containing approximately 25% ME from MCT oil and 25% ME from butter. Following a 12 hour fast, baseline blood samples were collected, and each dog was fed a test diet. Blood sampling was repeated at 30, 120, and 180 minutes post-prandially. Following a three to four-day wash-out period on the regular maintenance diet (Purina ONE® Lamb & Rice Formula Dry Dog Food, Nestlé Purina PetCare Company, St. Louis, MO), each dog was assigned a new treatment diet, and the process was repeated until all dogs received all treatments.

### Experimental subjects

Following approval by the University of Tennessee's Institutional Animal Care and Use Committee (IACUC 2862), 12 adult purpose-bred research beagles were enrolled into the study. All dogs were determined to be healthy upon physical examination and received a complete blood count, chemistry panel, and urinalysis within the previous 12 months. Each dog was randomly assigned to one of four dietary treatment groups by blindly drawing each dog's name from a bowl.

### Diet design

All experimental diets were made with a base of Pacific cod skinless filets (Great Value, Bentonville, AR), Mahatma extra-long enriched rice (Riviana, Houston, TX), and fat free Swanson® chicken broth (Campbell Soup Company, Camden, NJ). The cod and rice were prepared based on the manufacturer's directions, and then ingredients were combined and

frozen. Fat treatments and broth were added prior to feeding. The LCSFA diet contained 50% ME from butter (Land O'Lakes, Saint Paul, MN). The LCUFA diet contained 50% ME from canola oil (Great Value, Bentonville, AR). The high MCT diet contained 25% ME from MCT oil (NOW®, Bloomingdale, IL) and 25% ME from butter. The MCT treatment group contained both MCT and butter due to palatability issues when feeding 50% ME from MCT oil. Dietary nutrient profiles of each of the diets are listed in Table 1.

Test meals provided 50% of a dog's daily energy requirements (DER) to mimic a twice-daily feeding regimen. The DER of each dog was calculated based on the formula:130*[BW (kg)^0.75] [15]. While the control diet had a high proportion of protein and carbohydrate compared to test diets, this diet was fed at 25% DER to provide equal total intake of these nutrients with test diets. Therefore, all diet groups provided similar intake of protein and carbohydrates in the meal and differed only in fatty acid quantity or type.

## Sample collection and analysis

Jugular blood samples were collected into 2 mL vacutainer serum clot activator tubes (Greiner Bio-One North America Inc. Monroe, NC). Samples were immediately placed on ice and centrifuged in a refrigerated centrifuge at 3000x for 10 minutes to separate the serum. The serum was stored in cryogenic vials (Premium Vials, Tullytown, PA) and frozen at -80°C before being shipped on dry ice to the Texas A&M University (TAMU) Gastrointestinal Laboratory for sample analysis. The cPLI was measured using Idexx Spec PL ELISA kit (IDEXX Laboratories, Inc., Westbrook, ME). Gastrin measurement was conducted using Siemens Immulite XPi (Siemens Medical Solutions USA Inc. Malvern, PA). DGGR lipase was tested using Sentinel Lipase NG assay (Via Robert Koch, 2–20152 Milano, Italy). Amylase, triglycerides, and cholesterol were measured using Beckman Coulter reagents (Beckman Coulter Inc., 250 South Kraemer Boulevard, Brea, CA). DGGR lipase, amylase, triglycerides, and cholesterol were run on a Beckman AU480 (Beckman Coulter Inc., 250 South Kraemer Boulevard, Brea, CA).

CCK measurements occurred at the primary institution using a canine-specific ELISA (ELISA kit 2885300, MyBioSource, San Diego, CA). While the manufacturer-reported sensitivity was <4.0pg/mL, intra-assay variation ≤5.6%, inter-assay variation ≤7.9%, independent validation of the assay was beyond the financial scope of this project. Results from the CCK assay were not consistent with physiologic values previously reported for dogs [16, 17] and were below the sensitivity for the assay. As such the results were deemed questionable and not reported.

## Statistical analysis

A power analysis was performed using PASS 2021 (Power Analysis and Sample Size Software (2021). NCSS, LLC. Kaysville, Utah, USA, ncss.com/software/pass). Based on detecting a two-

**Table 1. Nutrient profile of diets (grams/1000Kcal).**

| Nutrient | Control | LCSFA | LCUFA | High MCT |
|---|---|---|---|---|
| Protein | 116 | 60 | 60 | 60 |
| Total fat | 4 | 57 | 57 | 56 |
| Carbohydrate | 115 | 59 | 59 | 60 |
| PUFA | 1.4 | 2.7 | 16.1 | 1.7 |
| MUFA | 0.8 | 16.2 | 35.1 | 8.1 |
| MCT | 0 | 5.6 | 0 | 31.5 |
| Saturated FA | 0.8 | 35.2 | 4.5 | 44.5 |

fold difference in cPLI, a four-treatment cross-over design with a sample of 12 subjects measured at four time points achieves 80% power to detect treatment differences among the means at a 0.05 significance level. The standard deviation across subjects for cPLI (primary aim) at the same time point was assumed to be 75 ng/mL. The pattern of the covariance matrix was to have all correlations equal with a correlation of 0.50 between the first and second time point measurements.

Repeated measures mixed model ANOVA as a crossover design, using a Kronecker product unstructured covariance matrix, was applied to test the within subject effects of treatment and time on cPLI, gastrin, amylase, cholesterol, triglycerides, and DGGR lipase. Ranked transformation was applied to cPLI and DGGR lipase to address non-normal distribution of the residuals. Post-hoc pairwise comparisons were conducted using Tukey-Kramer adjusted p-values. Shapiro-Wilk tests for normality were used to evaluate normality of the residuals. All statistical assumptions were sufficiently met. Statistical analysis was performed using SAS (version 9.4, Cary, North Carolina 27 513, USA, Release TS1M7). Statistical significance was set at $p < .05$.

## Results

A total of 14 dogs were enrolled into the study. One dog was removed due to diarrhea that occurred while eating the regular maintenance diet. Another dog was removed for refusing to eat the MCT diet. No data from these dogs were included in the final analysis. The mean body weight, body condition score (BCS, 9-point scale), and age of the remaining 12 dogs were 12.14 kg +/- 7.5, 6/9 +/- 3, and 5 years +/- 2, respectively. There were eight neutered males and four spayed females. All values are reported as mean +/- SEM.

There were no treatment effects on serum cPLI (Fig 1), gastrin (Fig 2), amylase (Fig 3), DGGR lipase (Fig 4), or cholesterol (Fig 5). However, serum triglyceride (Fig 6) concentrations had significant treatment by time interaction ($p < .0001$). Error bars represent mean +/- SEM in all figures. When looking within each timepoint, triglyceride concentrations differed significantly among groups at both the 120 minute and 180 minute timepoints ($p < .0001$) but not at the 0 minute timepoint ($p = .75$) or 30 minute timepoint ($p = .35$). Triglyceride concentrations reached the highest concentration at 120 minutes post-meal for LCSFA (154.7 mg/dL +/- 11.4)

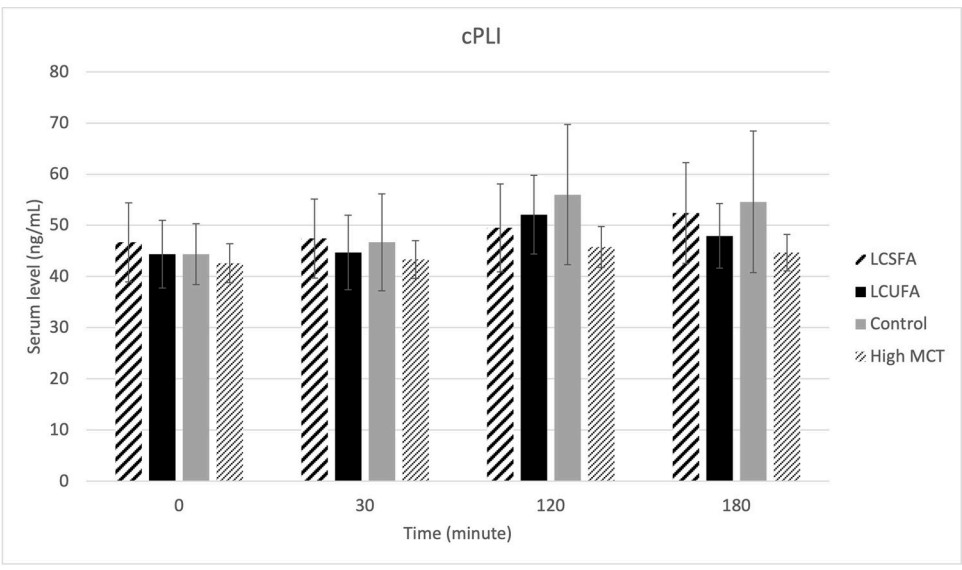

**Fig 1. Serum PLI.** Serum cPLI level before food intake (0 min), and 30min, 120min, 180min post-prandial.

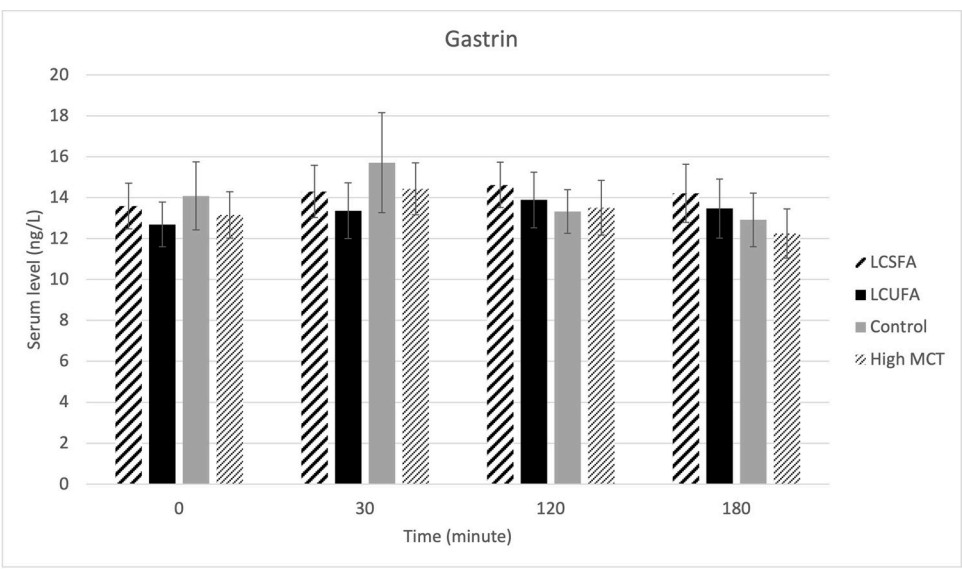

**Fig 2. Serum gastrin.** Serum gastrin level before food intake (0 min), and 30min, 120min, 180min post-prandial.

and LCUFA (114.2 mg/dL +/- 10.1), with the largest change from baseline noted at 120 min time point, and no significant changes in concentration between 120 to 180 minutes. The high MCT group had significantly lower triglyceride levels (p < .0001) compare to the long-chain fatty acid groups at 120 minutes (58.9 mg/dL +/- 4.9), and reached to higher concentration (p < .0001) at 180-minute timepoint (76.8 mg/dL+/- 18.5). As expected, the low-fat control diet had the lowest 120-minute triglyceride concentration at 43.2 mg/dL +/- 3.1. The full datasets for serum cPLI, gastrin, amylase, DGGR lipase, cholesterol, and triglycerides can be found in the S1 File.

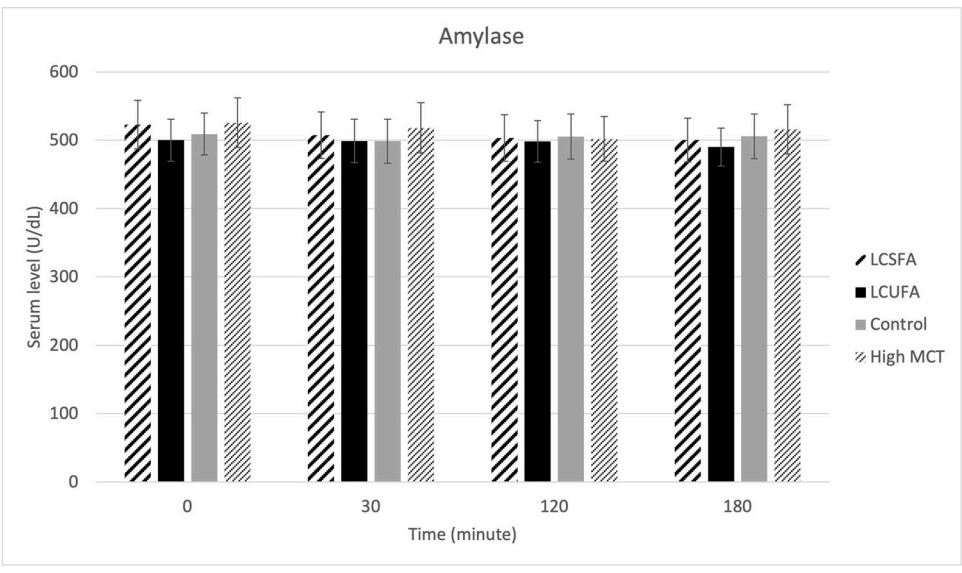

**Fig 3. Serum amylase.** Serum amylase level before food intake (0 min), and 30min, 120min, 180min post-prandial.

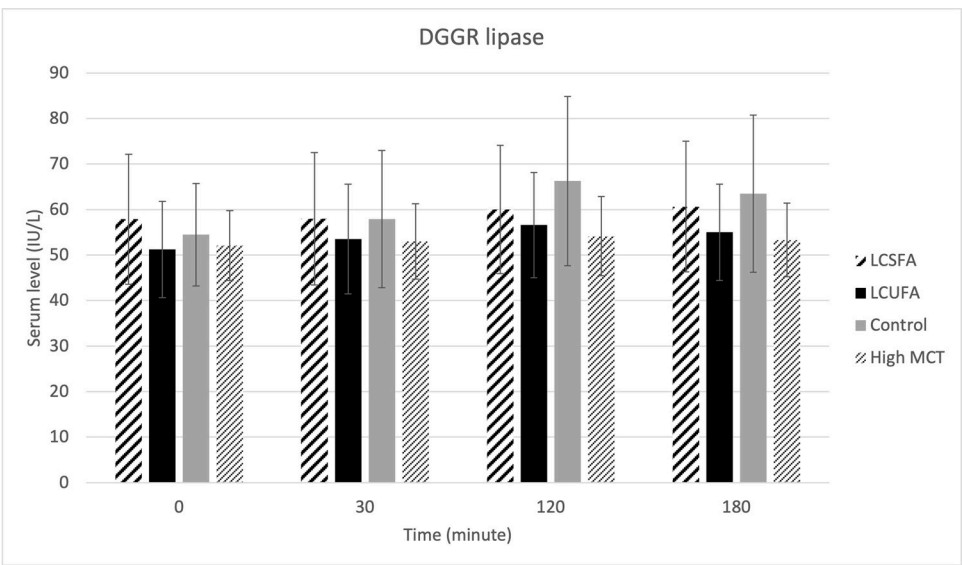

**Fig 4. Serum DGGR lipase.** Serum DGGR lipase level before food intake (0 min), and 30min, 120min, 180min post-prandial.

## Discussion

In human studies, medium chain triglycerides (MCT) have been shown to be more rapidly absorbed and result in less pancreatic enzyme release compared to long chain fatty acids [11]. The goal of this study was to determine if this same effect occurs in dogs by measuring markers of exocrine pancreatic stimulation. Based on our results, the replacement of LCFA with MCT has little effect on pancreatic lipase release in healthy dogs. Although MCT were expected to be absorbed in the small intestine with minimal release of pancreatic lipase, the present

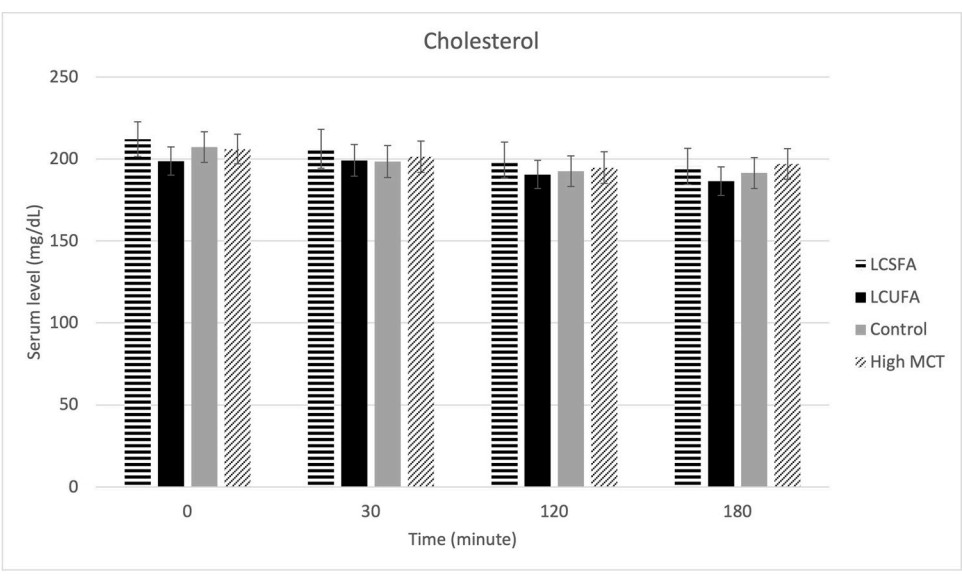

**Fig 5. Serum cholesterol.** Serum cholesterol level before food intake (0 min, and 30min, 120min, 180min post-prandial.

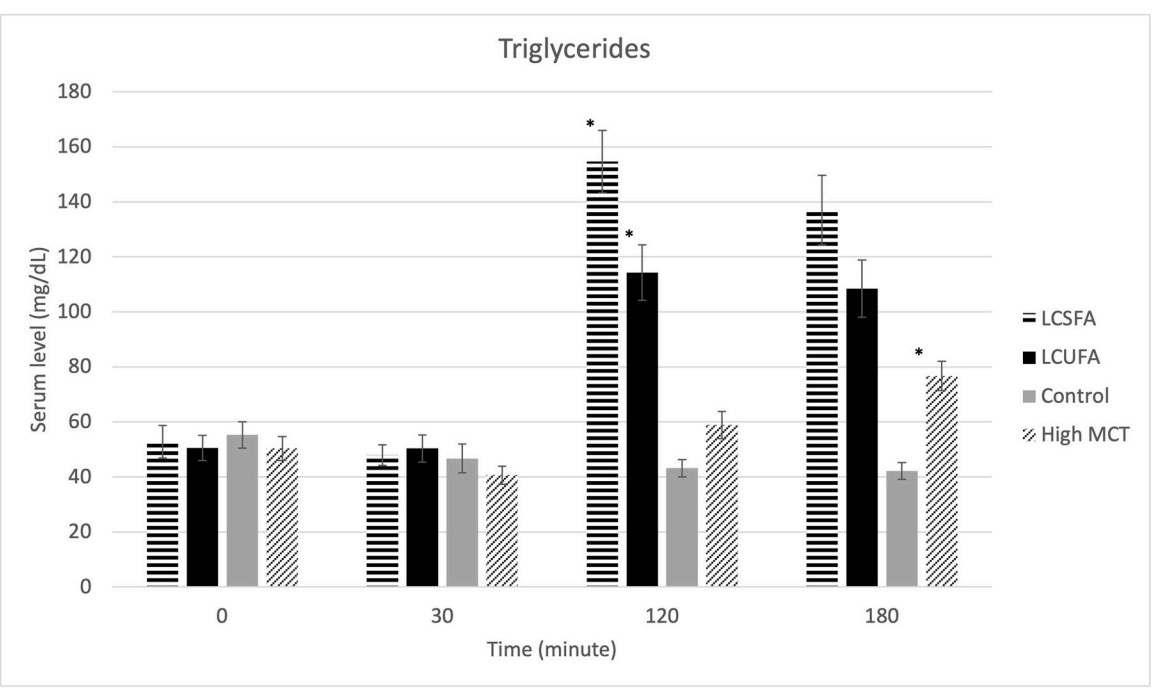

**Fig 6. Serum triglycerides.** Serum triglycerides level before food intake (0 min), and 30min, 120min, 180min post-prandial, the *
indicates that the level is different (p<0.001) than the previous time point.

experiment had to use a combination of LCSFA and MCT in the high MCT group due to the
poor palatability of MCT alone. The addition of LCFA may have produced enough pancreatic
lipase release to confound the effects of MCT. In addition, this study used healthy dogs with
presumably normal pancreatic function. The influence of dietary fat may be less pronounced
in this population, and our results are similar to a previous study of healthy dogs supplemented
with MCT [14]. Despite high fatty acid intake, the healthy pancreas can release appropriate
amounts of enzymes for normal fat digestion. Alterations in the dietary fatty acid composition
in dogs with chronic pancreatitis may yield different results.

Considering cPLI had no significant changes, it is unsurprising that amylase also demon-
strated minimal change. Changes in amylase activity tend to parallel the pancreatic lipase activ-
ity in dogs with pancreatitis [18]. In addition, gastrin and amylase secretion is not stimulated
by fatty acids alone and may have been impacted by the protein and carbohydrates within the
test diets.

The only marker found to have significantly changed among different test diets was the
serum triglyceride concentrations. Within the gastrointestinal tract, MCT and LCT are
digested to their respective fatty acids. Following absorption into the enterocyte, products of
LCFA digestion are esterified to triglycerides and repackaged into chylomicrons for transport
via the lymphatic system to the peripheral circulation. A diet high in LCFA is expected to raise
post-prandial serum triglyceride concentrations as the concentration of triglyceride-filled chy-
lomicrons increases. In contrast, because of their shorter chain lengths, medium-chain fatty
acids (MCFA) do not require chylomicron formation for their absorption and most MCFA
are transported and travel directly to the liver via the portal circulation [19]. The lower post-
prandial triglyceride concentrations of the high MCT group found in this study agrees with a
similar rodent study [20]. However, the effects of MCT on lowering serum triglycerides in
humans are less clear [21]. The lower concentration of post-prandial serum triglycerides in the

MCT group could prove beneficial for dogs with hypertriglyceridemia and further research is warranted in this specific population.

## Conclusion

Using MCT compared to long chain saturated or unsaturated fatty acids as the fatty acid source reduced post-prandial triglyceride levels in dogs, which might prove helpful in managing hyperlipidemia and hypertriglyceridemia in dogs. MCT supplementation does not appear to have a significant effect on pancreatic enzyme release in healthy dogs. Future research testing the effects of MCT on markers of pancreatic enzyme release, including CCK, in dogs with clinical pancreatitis is needed.

## Supporting information

**S1 File. Raw datasets for serum cPLI, gastrin, amylase, DGGR lipase, cholesterol, and triglycerides.**
(PDF)

## Acknowledgments

We would like to thank the faculty and staff of the Texas A&M University Gastrointestinal Laboratory for their generosity of time and equipment to analyze serum samples for this project.

## Author Contributions

**Conceptualization:** Yunyi Zhang, Claudia A. Kirk, M. Katherine Tolbert, Jörg M. Steiner, Dallas Donohoe, Maryanne Murphy, Angela Witzel-Rollins.

**Data curation:** Yunyi Zhang, Claudia A. Kirk, Cary Springer, Angela Witzel-Rollins.

**Formal analysis:** Yunyi Zhang, Claudia A. Kirk, Cary Springer, Angela Witzel-Rollins.

**Funding acquisition:** Yunyi Zhang, Angela Witzel-Rollins.

**Investigation:** Yunyi Zhang, M. Katherine Tolbert, Jörg M. Steiner, Angela Witzel-Rollins.

**Methodology:** Dallas Donohoe, Maryanne Murphy, Angela Witzel-Rollins.

**Project administration:** Yunyi Zhang, Angela Witzel-Rollins.

**Software:** Cary Springer.

**Supervision:** Claudia A. Kirk, Dallas Donohoe, Maryanne Murphy, Angela Witzel-Rollins.

**Writing – original draft:** Yunyi Zhang.

**Writing – review & editing:** Claudia A. Kirk, M. Katherine Tolbert, Jörg M. Steiner, Dallas Donohoe, Maryanne Murphy, Cary Springer, Angela Witzel-Rollins.

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
