## [Decision Letter · Decision Letter 0]

27 Jun 2023

PONE-D-23-14321Impact of fatty acid composition on markers of exocrine pancreatic stimulation in dogsPLOS ONE

Dear Dr. Witzel Rollins,

Thank you for submitting your manuscript to PLOS ONE. After careful consideration, we feel that it has merit but does not fully meet PLOS ONE’s publication criteria as it currently stands. Therefore, we invite you to submit a revised version of the manuscript that addresses the points raised during the review process. Please submit your revised manuscript by Aug 11 2023 11:59PM. If you will need more time than this to complete your revisions, please reply to this message or contact the journal office at plosone@plos.org. Please include the following items when submitting your revised manuscript:A rebuttal letter that responds to each point raised by the academic editor and reviewer(s). You should upload this letter as a separate file labeled 'Response to Reviewers'.A marked-up copy of your manuscript that highlights changes made to the original version. You should upload this as a separate file labeled 'Revised Manuscript with Track Changes'.An unmarked version of your revised paper without tracked changes. You should upload this as a separate file labeled 'Manuscript'.If applicable, we recommend that you deposit your laboratory protocols in protocols.io to enhance the reproducibility of your results. Protocols.io assigns your protocol its own identifier (DOI) so that it can be cited independently in the future. For instructions see: https://journals.plos.org/plosone/s/submission-guidelines#loc-laboratory-protocols. Additionally, PLOS ONE offers an option for publishing peer-reviewed Lab Protocol articles, which describe protocols hosted on protocols.io. Read more information on sharing protocols at https://plos.org/protocols?utm_medium=editorial-email&utm_source=authorletters&utm_campaign=protocols.

We look forward to receiving your revised manuscript.

Kind regards,

Ewa Tomaszewska, DVM Ph.D

Academic Editor

PLOS ONE

Journal Requirements:

   "Funded through intramural University of Tennessee research grant."

Reviewers' comments:

Reviewer's Responses to Questions

**Comments to the Author**

1. Is the manuscript technically sound, and do the data support the conclusions?

Reviewer #1: Yes

2. Has the statistical analysis been performed appropriately and rigorously? 

Reviewer #1: Yes

3. Have the authors made all data underlying the findings in their manuscript fully available?

Reviewer #1: Yes

4. Is the manuscript presented in an intelligible fashion and written in standard English?

Reviewer #1: Yes

5. Review Comments to the Author

Reviewer #1: This manuscript is well written, the design and measurements are understandable and is a pleasure to read. I only have minor textual comments.

Specific comments:

L93: This statement needs a reference.

L127: Why mentioning serum CCK, gastrin, and cPLI, but not the other markers (e.g. amylase etc.)

Legends to the Figures: error bars shown represent presumable mean ± SD (according the supplementary support, but this must be clearly stated in the manuscript. Suggest to do this in all legends (or under Statistical Analysis).

L228-231: Give P values for “The high MCT group had … at 43.2 mg/dL +/- 3.1.”

L233-242: Suggest to mention 0 min in the legend, like “…before food intake (0 min) and 30 min….”

6. PLOS authors have the option to publish the peer review history of their article (what does this mean?). If published, this will include your full peer review and any attached files.

Reviewer #1: No

---

## [Author Response · Author response to Decision Letter 0]

26 Jul 2023

We would like to thank the reviewer for their comments to make our research finding more clear to the reader. We have addressed your concerns as follows:

L93: Reference has been added.

L127: Other markers were listed and a description of the error bars was added to the statistical analysis section.

L228-L231: P values were added

L233-242: 0 min has been added to the food intake time period.

---

## [Editor Report · Decision Letter 1]

10 Aug 2023

Impact of fatty acid composition on markers of exocrine pancreatic stimulation in dogs

PONE-D-23-14321R1

Dear Dr. Angela Witzel Rollins,

We’re pleased to inform you that your manuscript has been judged scientifically suitable for publication and will be formally accepted for publication once it meets all outstanding technical requirements.

Kind regards,

Ewa Tomaszewska, DVM Ph.D

Academic Editor

PLOS ONE
---

## [Editor Report · Acceptance letter]

16 Aug 2023

PONE-D-23-14321R1 

Impact of fatty acid composition on markers of exocrine pancreatic stimulation in dogs 

Dear Dr. Witzel-Rollins:

I'm pleased to inform you that your manuscript has been deemed suitable for publication in PLOS ONE. Congratulations! Your manuscript is now with our production department. 

Kind regards, 

on behalf of

Professor Ewa Tomaszewska 

Academic Editor

PLOS ONE